# Recognition of Maize Phenology in Sentinel Images with Machine Learning

**DOI:** 10.3390/s22010094

**Published:** 2021-12-24

**Authors:** Alvaro Murguia-Cozar, Antonia Macedo-Cruz, Demetrio Salvador Fernandez-Reynoso, Jorge Arturo Salgado Transito

**Affiliations:** 1Colegio de Postgraduados, Campus Montecillo, Carretera Federal Mexico-Texcoco, km. 36.5, Montecillo, Texcoco 56230, State of Mexico, Mexico; murguia.alvaro@colpos.mx (A.M.-C.); demetrio@colpos.mx (D.S.F.-R.); 2Colegio Mexicano de Especialistas en Recursos Naturales AC, De las Flores no. 8 s/n, San Luis Huexotla, Texcoco 56220, State of Mexico, Mexico; arturo.transito@gmail.com

**Keywords:** support vector machine, local indicator of spatial association, local binary pattern, texture characteristic, colour characteristic, leaf area index

## Abstract

The scarcity of water for agricultural use is a serious problem that has increased due to intense droughts, poor management, and deficiencies in the distribution and application of the resource. The monitoring of crops through satellite image processing and the application of machine learning algorithms are technological strategies with which developed countries tend to implement better public policies regarding the efficient use of water. The purpose of this research was to determine the main indicators and characteristics that allow us to discriminate the phenological stages of maize crops (*Zea mays* L.) in Sentinel 2 satellite images through supervised classification models. The training data were obtained by monitoring cultivated plots during an agricultural cycle. Indicators and characteristics were extracted from 41 Sentinel 2 images acquired during the monitoring dates. With these images, indicators of texture, vegetation, and colour were calculated to train three supervised classifiers: linear discriminant (LD), support vector machine (SVM), and k-nearest neighbours (kNN) models. It was found that 45 of the 86 characteristics extracted contributed to maximizing the accuracy by stage of development and the overall accuracy of the trained classification models. The characteristics of the Moran’s *I* local indicator of spatial association (LISA) improved the accuracy of the classifiers when applied to the L*a*b* colour model and to the near-infrared (NIR) band. The local binary pattern (LBP) increased the accuracy of the classification when applied to the red, green, blue (RGB) and NIR bands. The colour ratios, leaf area index (LAI), RGB colour model, L*a*b* colour space, LISA, and LBP extracted the most important intrinsic characteristics of maize crops with regard to classifying the phenological stages of the maize cultivation. The quadratic SVM model was the best classifier of maize crop phenology, with an overall accuracy of 82.3%.

## 1. Introduction

The mismanagement of water resources and deficiencies in their distribution and application are serious problems that increase the volume of water necessary for the establishment of crops, which has a negative impact on the availability of water resources and results in large economic losses in the agricultural sector. Population growth demands a greater quantity of food, which implies producing more in the same area, for which water is a determining factor [1]. The modernization of hydro-agricultural infrastructure and the application of more efficient irrigation systems are actions that are being applied in several countries of the world to increase irrigation efficiency and improve water management. González-Trinidad et al. [2] evaluated water efficiency and agricultural productivity in a modernized semi-arid region in central-north Mexico. Their results indicated a 30% increase in global efficiency and higher yields in all crops. Gonçalves et al. [3] conducted an efficiency analysis in the Lis Valley Irrigation District in Portugal and found that the overall efficiency in the district was 67% and concluded that it is important to maintain, conserve and modernize the hydraulic infrastructure. In Spain, for three decades, they have promoted the use of pressurized irrigation systems to reach 72% of the arable area [4], in Brazil, the main modern irrigation technique is the central pivot irrigation system with a 78.3% of irrigable area [5].

These actions in conjunction with the applying irrigation according to the needs of the crop impacts the saving of water resources; however, this is limited by the lack of a model that efficiently determines the type of crop and the conditions of the plant in a specific region. One option for the generation of such information is the processing of images acquired from remote sensors with classification models based on artificial vision. These can be automatic (unsupervised) and/or supervised; unsupervised methods apply grouping algorithms through the determination of thresholds [6], and supervised algorithms use pattern recognition based on datasets with labelled characteristics for the training and testing of the model.

There are studies referring to the identification of vegetation cover through image segmentation. Macedo et al. [7] applied a modification to the Otsu thresholding method to classify six types of vegetation cover. Oliva et al. [8] proposed a segmentation algorithm based on the 3D histogram of intensities and the Gauss–Markov measurement field model for the categorization of eight crops of economic importance. Wu et al. [9] designed an adaptive segmentation algorithm calculated from Euclidean distances. García-Martínez et al. [10] calculated the plant cover fraction in maize (*Zea mays* L.) with grouping algorithms applied to four vegetation indices: the triangular greenness index (TGI), excess green index (EXG), visible atmospheric resistance index (VARI) and normalized green–red difference index (NGRDI) from seasonal images acquired by unmanned aerial vehicles (UAVs).

Supervised classification models such as convolutional neural networks [11], decision trees [12], vector support machines [13,14], some hybrid models that combine the segmentation of spatial dependence data with supervised classifiers [15], and clustering algorithms and artificial neural networks [16] have also been applied. D’Adrimont et al. [17] trained supervised decision tree classifiers with a normalized difference yellow index (NDYI) time series obtained from Sentinel 1 and Sentinel 2 images to determine the flowering dates of rapeseed crops (*Brassica napus*).

In recent years, the use of data extracted from Sentinel 1 and 2 satellite images for the classification of agricultural crops has increased [18,19,20,21,22,23,24,25,26]. Hejmanowska et al. [18] used a limited number of Sentinel 1 and 2 images to classify established crops in central Poland, by training a Random Forest (RF) classification model; the precisions reported by the authors were 81% when data extracted from Sentinels 1 and 2 were combined and 79% when only Sentinel 2 data were used. Rao et al. [19] discriminated corn, mustard, tobacco, and wheat crops using a support vector machine model (SVM) trained from Sentinel 1 and 2 data with an accuracy of 85%. Saini and Ghosh [20] trained two RF and SVM classification models to discriminate 11 soil covers with accuracies of 84.22% and 81.85%, respectively. Brinkhoff et al. [21] generated coverage maps of nine perennial crops and three classes (annual crops, forest crops and others) from a SVM model trained with Sentiel 2 data and synthetic aperture radar images, which reached a precision of 84.8%. Belgiu and Csillik [22] implemented the time-weighted dynamic time warping (TWDTW) model based on pixels and objects extracted from Sentinel 2 images, to classify various agricultural crops in three study areas. The precisions achieved by the classifier were 78.05% for pixels and 96.19% objects. Maponya et al. [23] compared the SVM classification models, decision trees (DT), k-Nearest neighbors (kNN), RF and Maximum Likelihood (ML) to discriminate canola, lucerne, pasture, wheat, and fallow, based on Sentinel 2 images; SVM and RF models stood out with 77.2%. Neetu and Ray [25] evaluated the RF models, classification, and regression trees (CART) and SVM to classify crops from data extracted from Sentinel 2 images. The CART model was the best with 93.3% and a kappa coefficient of 0.9178. Yun et al. [26] proposed a multilevel classification model based on Sentinel 1, radar, and Sentinel 2 data to recognize six crops with an accuracy of 98.07%.

The determination of the phenological stages of a crop is a complex process because there are different factors that influence the performance of the classification algorithms, so it is essential to look for indicators that minimize the noise caused by these factors. Several research studies have published positive results regarding the identification of the phenological stages of crops through high-resolution images taken with conventional cameras using different indicators. In this sense, Yalcin [27] trained a naïve Bayes classifier with the grey level cooccurrence matrix (GLCM) and histogram of oriented gradients (HOG) indicators to classify the phenological stages of crops of wheat, barley, lentil, cotton, pepper, and maize from images taken with cameras of mobile devices (cell phones and tablets) for the Turkish Agricultural Monitoring and Information System (TARBIL). Likewise, Zheng et al. [28] performed a time series analysis of vegetation indices to determine the phenological stages of rice cultivation from canopy spectra captured by two portable spectrometers. On the other hand, Han et al. [29] and Yang et al. [30] used texture and colour indicators to discriminate the main phenological stages of rice cultivation; Han used images taken with conventional cameras, while Yang used images taken with cameras mounted on drones. A large number of studies related to the recognition of a phenological stage, or the complete phenology of a specific agricultural crop are based on the processing of digital images acquired with handheld cameras or drones. These images are characterized by a high spatial resolution and an absence of cloudiness, but the area that can be studied is smaller than that of satellite images. The cost that the use of this type of images would imply is very high compared to Sentinel 2 images, coupled with the near infrared (NIR) band that Sentinel provides and that conventional cameras do not have. In addition, the high energy consumption of drones limits the flight time and with it the surface that can be monitored in one day.

The purpose of this research was to recognize the phenological stages of a maize crop (*Zea mays* L.) in Sentinel 2 satellite images through supervised classification models. For this purpose, 148 plots cultivated with maize with different planting dates were monitored during an agricultural cycle. The particular objectives of this research were to determine the indicators representing the intrinsic characteristics of maize crops that allow discrimination of its phenology (indicators of texture, vegetation, and colour). Three supervised classification models (linear discriminant (LD), SVM and kNN models) were compared, and their performance based on the extracted characteristics was evaluated.

The hypothesis of this research was that the greater the number of characteristics, the greater the precision of the supervised learning classifier in determining the phenology of maize crops from satellite images. The main contributions of this study were as follows: the characteristics that best identify the phenology of the maize crop were determined through the analysis of Sentinel 2 satellite images. Three supervised recognition models were trained to determine the phenological stages of the maize crop, and the classification models with different groups of characteristics were evaluated. The Moran’s *I* local indicator of spatial association (LISA) improved the accuracy of the classifiers when applied to the L*a*b* colour model and to the red, green, blue, (RGB) and near infrared (NIR) bands.

## 2. Materials and Methods

### 2.1. Location and Monitoring of the Control Plots

The research was developed in the Irrigation Module “Tepatepec”, Irrigation District 003 “Tula” in the state of Hidalgo, located between the parallels 20°12′ and 20°16′ north latitude and the meridians 99°00′ and 99°08′ west longitude, at an altitude of 1985 m (Figure 1). A total of 148 sampling plots were selected from irrigation sections 30, 31, 32, 33 and 34. Monitoring of the plots was carried out during the spring–summer 2019 agricultural cycle between March and November. Note that not all maize parcels in the district were planted on the same dates, so the date of planting of the crop ranged between 15 March and 15 May.

The development of the crops established in the sampling plots was monitored every week during the agricultural cycle. Visits to the plots were scheduled in such a way that they coincided with the passage of the Sentinel Satellite so that the plots could be correlated with the satellite data. During the monitoring, panoramic, point, and overhead photographs were taken of each analysed parcel (Figure 2), and the latter were taken at the same height when conditions allowed. In addition to the photographs, observations related to the conditions of the plant were recorded to identify the phenological stages and to build the spatiotemporal database.

### 2.2. Obtaining Satellite Images

The Sentinel 2 satellite images are free products with spatial resolutions of 10, 20 and 60 metres provided by the European Space Agency (ESA) and the European Commission through the Copernicus programme. The images are available in 13 different spectral bands. In the present study, bands 02, 03, 04, and 08 were used, corresponding to the RGB and NIR bands, all with a spatial resolution of 10 m. Forty-one satellite images of the study area with less than 50% cloud cover were studied, Table 1 presents more information about the characteristics of the satellite images used and the list sentinel 2 satellite images analyzed in this research work are available online as Appendix A. No pre-processing of the satellite images was carried out and the raw data of the analysed images was used.

### 2.3. Information Processing

#### 2.3.1. Geographic Information System

A geographic information system (GIS) was generated to gather the information and to generate the spatiotemporal database (plots, monitoring dates, photographs, and satellite images) to manage and analyse the data obtained during the monitoring through a relational scheme between the 41 satellite images and 148 sampling plots monitored during the 9 months of the 2019 agricultural cycle. From this information, the phenological stages of maize (*Zea mays* L.), taking as a reference the extended German phenological scale of cultivated plants *Bundesanstalt, Bundessortenamt and CHemical* (BBCH), are described below [31,32]:-Stage 1 “Emergence”: the plants have up to three unfolded leaves, with an average size of up to 15 cm and a height of 0.20 m.-Stage 2 “Development”: longitudinal growth of the main stem of the plant from 0.2 m to 2 m on average.-Stage 3 “Tassels and ears”: appearance of male (tassel) and female (ear) inflorescences in plants.-Stage 4 “Formation and maturation of the ear”: the cob has been completely formed, and the kernels have a milky consistency.-Stage 5 “Beginning of senescence”: the kernels are hardened and shiny, 65% dry matter has been reached, 0–50% dry plants.-Stage 6 “End of senescence”: plants 50–100% dry, ears ready to be harvested.

In the Figure 3, a map of the selected sampling plots displayed according to irrigation section is shown; the total area sampled amounts to 292.93 hectares distributed in 148 polygons. The plots that comprise the “Tepatepec” irrigation module are dominated by a network of distribution channels that divide their surface into five main irrigation sections, these sections allow users to correctly operate the hydro-agricultural infrastructure and distribute water to all plots.

#### 2.3.2. Extraction of Plots for Six Phenological Stages

The Sentinel 2 images were cut to a size of 804 × 1498 pixels, and spectral bands 02, 03, 04, and 08 were joined to form four-band compositions. Then, the pixels corresponding to the sample plots were extracted from the compositions and stored as files under the naming scheme *date composition_plot number_phenological stage*.

### 2.4. Calculation of the Indicators per Sample

Different indicators of texture, colour and vegetation were calculated for all the parcels analysed using algorithms programmed in MATLAB. Texture is a repeating pattern of intensities in an image, too fine to be distinguished as different objects at a certain spatial resolution [23]. Vegetation indexes are those that evaluate the conditions of the vegetation present on the earth’s surface, which are estimated through the relationship of the visible and near infrared spectrum bands. The colour indicators are determined by transforming the original data to other colour models (Section 2.4.4 and Section 2.4.5). The indicators used were Moran’s *I*, the local binary pattern (LBP), the leaf area index, colour models (RGB, hue—saturation—value (HSV), L*a*b* and YIQ) and RGB colour ratios.

#### 2.4.1. Moran’s *I* Local Indicator of Spatial Association

Moran’s *I* is a LISA determining the spatial association between the same variable in two adjacent locations. Applying this concept to the processing of satellite images allows the calculation of the correlation that exists between neighbouring pixels, which can be used as a textural characteristic in the design of classifiers [33]. The formula to estimate this LISA is presented in Equation (1).
(1)$$I_{i}=\frac{x_{i}-\bar{X}}{S_{i}^{2}}\overset{n}{\underset{j=1,\;\;j\neq i}{\sum }}w_{ij}\left(x_{j}-\bar{X}\right)$$
where

*x_i_* analysed pixel$\bar{X}$ global mean$S_{i}^{2}$ global variance*x_j_* neighbour pixel$w_{ij}$ type W, C, U, B weight matrices*n* total number of pixels in the image

The matrix of the relative weights *w* is a square matrix *nxn* that assigns a weight value to the eight neighbours of the pixels contained in the analysed image and zero for the non-neighbouring pixels. The weights of the weight matrix W are calculated as 1/number of neighbours per pixel, and the sum of the rows of the matrix W is equal to 1. C is obtained as n/global number of neighbourhoods, and the sum of the weights of matrix C is equal to n. U is estimated as 1/global number of neighbourhoods, and the sum of the weights of matrix U is equal to 1. Finally, the type B matrix is equal to 1 for the neighbouring pixels and 0 for non-neighbours.

#### 2.4.2. Local Binary Pattern

The LBP operator is a texture operator that calculates the value of a pixel as a function of its neighbourhoods. The *LBP* evaluates whether the intensities of the neighbours exceed a certain threshold and codes those that are greater than the threshold as 1 and those that are smaller as 0 [34]. Equation (2) presents the formula for estimating the indicator.
(2)$$LBP_{P,R}=\overset{P-1}{\underset{P=0}{\sum }}s\left(g_{P}-g_{c}\right)2^{P},\;\;\;\;\;\;s\left(x\right)=\{\begin{array}{c} 1\;\;\;\;\;si\;\;\;\;x\geq 0\\ 0\;\;\;\;\;si\;\;\;\;x<0 \end{array}$$
where *P* = 8 is the number of neighbours considered in the analysis, *R* is the size of the neighbourhood, and *g_p_* and *g_c_* are the values of the neighbouring and central pixels, respectively.

#### 2.4.3. Leaf Area Index

The leaf area index (*LAI*) is an indicator that determines the accumulated area of leaves per unit area; its values are dimensionless, and the higher the *LAI* is, the greater the photosynthetic production of a crop. Equation (3) presents the formula to calculate this indicator, which is based on the research of Bastiaanssen [35].
(3)$$LAI=-\frac{ln\left(\frac{0.69-SAVI}{0.59}\right)}{0.91}$$
where $SAVI=\frac{\left(1+L\right)\left(NIR-RED\right)}{L+NIR+RED}$ is the soil-adjusted vegetation index; *L* = 0.5, is a factor describing the density of vegetation which varies from −1 to 1, with 0 being dense vegetation and 1 being the low presence of vegetation; NIR is the near-infrared spectral band; and RED is the red spectral band.

#### 2.4.4. Colour Models

The models or colour spaces are methodologies of colour representation through numerical triads that represent, according to the model, values of primary colours, hue, saturation, chromaticity, brightness, or luminance. There are different types of colour models. In the present work, the RGB model; hue, saturation, value (HSV) model; L*a*b* model of the International Commission on Illumination (CIE, for its acronym in French) (1976), and luma in-phase quadrature (YIQ) model, which separates colour from brightness, were used [36].

#### 2.4.5. Colour Indicators

The colour ratios relate the spectral bands of the RGB colour model to determine the excess green in the digital images. These ratios have been used to identify diseases in the cultivation of alfalfa [37] to estimate the plant cover of maize [10] and to detect the phenology of rice cultivation [29]. Equation set (4) was used to calculate these ratios.
(4)$$r=\frac{R}{R+G+B};\;\;g=\frac{G}{R+G+B};\;b=\frac{B}{R+G+B}$$

### 2.5. Feature Extraction

Once the indicators (texture, colour, and vegetation) of all the samples were determined and programmed, an algorithm and its corresponding script for the extraction of characteristics were designed in MATLAB; this algorithm consists of importing the extracted samples (sets of pixels), applying the indicators, and calculating the mean and variance of each indicator/sample (characteristics of colour, texture, and vegetation), taking the plot as the region of interest.

The result of the designed algorithm was a matrix database with 4015 records (samples/rows) and 86 features (columns) made up of 26 colour features (2 features × 3 bands × 4 models + 2 features × 1 NIR), 52 texture features (2 features × 2 methods × 3 bands × 4 models + 2 features × 2 methods × 1 NIR), and 8 vegetation features (2 features × 3 bands + 2 features × 1 LAI).

### 2.6. Supervised Classification Models

#### 2.6.1. Linear Discriminant Classifier

The classifier based on the linear discriminant analysis applies the decision rule of a Bayes classifier and assumes that the density functions of each class are multivariate Gaussian. Equation (5) determines the classification probabilities of the linear discriminant (LD) model. This classification model has been used by Zheng at al. [38] to detect yellow rust in wheat crops from different vegetation indices extracted from Sentinel 2 satellite images and meteorological data. Borràs et al. [39] trained an LD model to discriminate 14 land covers with data extracted from Sentinel 2 and Spot images, achieving a precision of 72%. Peña et al. [40] analysed the time series of nine Landsat-8 satellite images from the years 2014 and 2015 to design an LD model that discriminates seven fruit crops.
(5)$$\delta_{k}\left(x\right)=x^{T}\sum^{-1}\mu_{k}-\frac{1}{2}\mu_{k}^{T}\sum^{-1}\;\mu_{k}+\log\pi_{k}$$

Optimizing Equation (5) with $G\left(x\right)=argmax_{k}\delta_{k}\left(x\right)$, the maximum conditional probabilities per class are determined and the samples are classified according to the maximum probability obtained. The training process of linear discriminant classifier consists of determining the parameters of the Gaussian distributions with the training data set [38]:

πk=NkN, where *N_k_* is the number of samples of class-*k*
$$\mu_{k}=\sum_{g_{i}=k}x_{i}/N_{k}$$
$$\hat{\sum }\;=\sum_{k=1}^{K}\sum_{g_{i}=k}\left(x_{i}-\hat{\mu_{k}}\right){\left(x_{i}-\hat{\mu_{k}}\right)}^{T}/\left(N-K\right)$$

The test stage of the classifier is run with the training data set and the parameters of Gaussian distribution calculated during training of model.

#### 2.6.2. k-Nearest Neighbours

The nearest neighbours’ methods use their observations of the training set T as a space of inputs *x* to form $\hat{Y}$. Specifically, the k-nearest neighbours method adjusts $\hat{Y}$ as follows [41]:(6)$$\hat{Y}\left(x\right)=\frac{1}{k}\underset{x_{i}\in N_{k}\left(x\right)}{\sum }y_{i}.\;$$
where *N_k_*(*x*) is the neighborhood of x defined for the k-nearest points *x_i_* in the training set. Closeness implies a metric, which can be Euclidean, Manhattan, or Miknowsky distance. The model must find the *k* observations with the closest *x_i_* to *x* and average their responses [41]. Based on the distance, the analysed sample is classified in the class to which the majority of the neighbours belong [42].

This classifier has shown important results in the classification of the land cover from data extracted from Sentinel 2 satellite images, reaching classification accuracies of 94% [43]. Qadri et al. [44] trained a kNN to classify cotton crop varieties from digital images of leaves, from which the texture characteristics were extracted, to form the training set, achieving a classifier precision of 99%. Navarro et al. [45] found that RGB images can be classified with high accuracies when analysed with kNN classification models.

#### 2.6.3. Support Vector Machine

Support Vector Machines (SVM) constitute one of the most powerful supervised classification techniques, and a standard in the state-of-the-art today. The model is a bi-class classifier, although most of its implementations (LIBSVM, LIBLINEAR) execute multiclass classification [46].

The goal of an SVM is to infer a decision boundary in the feature space, so that the subsequent observations are automatically classified into one of the two groups defined by that boundary (hyperplane). The SVM tries to generate such hyperplane in order to maximize the separation between each of the groups [46]. Equation (7) determines the decision hyperplane of the SVM model [47].
(7)$$f\left(x\right)=\underset{i\in S}{\sum }\alpha_{i}y_{i}K\left(x_{i},x_{j}\right)+b.\;$$
where *K*(*x_i_, x_j_*) is the kernel function, S the set of samples, *α_i_ y b* are parameters of the model that must be adjusted. The most common kernels are:

Linear: *K*(*x_i_*, *x_j_*) = (*x_i_*, *x_j_*)Polynomial:
$K\left(x_{i},x_{j}\right)={\left(\gamma (x_{i},x_{j})+c\right)}^{d}$Radial basis function: $K\left(x_{i},x_{j}\right)=exp\left\{-\frac{\|x_{i}-x_{j}{\|}^{2}}{2\sigma^{2}}\right\}$
Sigmoid: *K*(*x_i_*, *x_j_*) = *tanh*(*γ*(*x_i_*, *x_j_*) + *c*)

The SVM model used in the present investigation was trained with a polynomial type of kernel function of degree two (quadratic SVM). SVM classifiers have been used in numerous applications, analysing remote sensing data. The main tasks that have been addressed to these kind of classification models are the monitoring and estimation of biophysical parameters, such as chlorophyll concentration, gross primary productivity and evapotranspiration; vegetation classification (forest density, forest species, forest degradation, forest cover changes, crop classification and semi-arid vegetation areas); the analysis of urban areas and impervious surfaces such as roads for their vectorization and vector layers elaboration, extraction of urban areas of interest and recognition of structures [48]. In recent years, different studies have been carried out on the recognition of crops and discrimination of different land covers from data extracted from Sentinel 2 satellite images, applied to the design of SVM classification models [19,20,21,23,25].

#### 2.6.4. Training and Validation of the Models

Based on the characteristics determined, three supervised classifiers were trained with the objective of discriminating six phenological stages of maize crops on 90% of the 4015 samples and the 86 characteristics extracted from the Sentinel 2 satellite images, leaving 10% of the samples for validation. The distribution of the samples by the phenological stage is shown in Table 2.

The training process was executed with the *Classification Learner* tool of MATLAB software version 2017. The supervised classification models chosen were LD, quadratic SVM, and kNN classifiers. To determine the accuracy of the models, *k-fold* cross-validation was used [49,50], as it helps prevent overfitting since it allows part of the available data to be saved and used as a test dataset.

For this validation method, *k* = 10; this the number of groups into which the dataset available for each phenological stage of maize crops is separated. Therefore, the dataset was divided into ten equal parts, 10 iterations were executed, where nine groups were randomly selected for training, and one was reserved for the validation of the models. In this way, the model was generated only with the training data. With the created model, the output data were generated in a confusion matrix and were compared with the set of data reserved for validation (not used in the training of the generated model).

The statistics used to evaluate the classifier included the overall precision of the model and the precision for each class (phenological stage of maize crops). Equation (8) allows us to determine the level of general precision reached by the classification model. Equation (9) allows us to determine the precision by class. These metrics are based on the results of the confusion matrix [46].
(8)$$Overall\;precision=\frac{TP}{TP+FP}$$
(9)$$\;Class\;precision=\frac{TP_{i}}{TP_{i}+FP_{i}}$$
where *TP* is the true positives, corresponding to the proportion of correctly classified samples; False positive *FP* is the false positives, the proportion of samples erroneously classified as positive; and *i* corresponds to each of the six established phenological stages.

#### 2.6.5. Debugging of the Detectable Phenological Stages

To reduce user error and to increase the accuracy of the classifiers, the samples corresponding to stage three (tassels and ears) were eliminated, and the distribution of the training samples was refined considering the transition dates between phenological phases (through analysis by an expert) of the sampling data, leaving the refined model for the determination of five phenological stages. From this debugging, 3764 samples were selected, and their phenological stages were distributed, as indicated in Table 3. With this information, the spatial-temporal database was updated, and the algorithm of extraction of the plots, descriptors and characteristics was executed for a second time.

In the second training phase, with the objective of discriminating five phenological stages with the supervised classifiers, 90% of the 3764 samples were used, with their 86 characteristics extracted from the Sentinel 2 satellite images, leaving 10% of the samples for validation. The *k-fold* cross-validation method called was again used [49,50], with k = 10, recording the results in a confusion matrix, which facilitated the calculation of statistics allowing the evaluation of the classifiers through the overall precision of the model and the precision by phenological stage of maize crops.

#### 2.6.6. Determination of the Most Representative Characteristics

In this research, the method used for the selection of characteristics was trial and error through the Classification Learner tool of MATLAB, which allows us to select, during the training, the best set of characteristics, facilitating the execution and validation of the models for the different combinations of characteristics.

The classifier was evaluated with the characteristics of the RGB colour model. Subsequently, more features were added, and the resulting precision of the new model was evaluated. If the precision decreased, then the characteristic was eliminated; otherwise, the characteristic was added to the model and to the list of representative characteristics. This process was repeated for all previously determined characteristics. At the end of the process, 45 representative characteristics were included. The supervised classification models were trained with the 45 characteristics selected and evaluated as representative.

#### 2.6.7. Evaluation of the Impact of the Group of Descriptors

Once the classification models were trained and validated with the 45 most representative characteristics of the descriptors of texture (LISA and LBP), vegetation (LAI and colour ratios) and colour (RGB and L*a*b*), the model with the greatest precision (quadratic SVM) was selected, and six classifiers were trained and validated.

For the first classifier of the 45 main characteristics, the LISA characteristics were omitted (Table 4, Row 2). In the second model, the LISA characteristics were re-added, and those of LBP were eliminated (Table 4, Row 3). For the third model, the LBP characteristics were re-added, and the LAI vegetation characteristics were omitted, as indicated in Table 4, Row 4. The same was performed for all six models, eliminating the characteristics corresponding to the indicators indicated in the last column of Table 4.

The hypothesis raised in this research was evaluated by comparing the performance of the classification models, as they were trained with the extracted characteristics, the most representative characteristics, and the impact of the descriptors.

## 3. Results and Discussion

In this study, a methodology was presented to classify the phenological stages of maize cultivation through the processing of Sentinel 2 satellite images and pattern recognition techniques. Texture, colour, and vegetation indicators were applied for the extraction of characteristics, and three supervised classification models were trained.

### 3.1. Training and Validation of the Supervised, Initial Classification Models

To recognize the six phenological stages of maize crops, three supervised classifiers LD, quadratic SVM, and kNN were trained using 86 characteristics extracted from Sentinel 2 satellite images with MATLAB Classification Learner software. In this stage, all the characteristics extracted from the samples were evaluated.

Table 5 shows the accuracies obtained with the trained classifiers by the phenological stage. The quadratic SVM model obtained the highest overall precision of 74.4%, although the tassel and ear stage could not be classified. This could be due to the limited set of samples due to the presence of cloudiness on the dates when this phenological stage occurred (June–July). It is not always possible to detect all phenological stages, as in the case of Han et al. [29], who trained an SVM classifier with colour and texture descriptors to determine the phenological stages in rice cultivation and found that the heading stage could not be discriminated by the model.

Figure 4 corresponds to the confusion matrix of the quadratic SVM model applied to the samples of the maize plots. From this, it is observed that all the stages, except for stages three “tassels and ears” and five “beginning of senescence”, could be classified with an accuracy greater than 70%. A total of 14% of the samples of stage one were classified as stage two, 8% of stage two as one, 11% of stage two as four, and 12% of stage four as two. Likewise, 21% of the samples from stage five were classified as stage four, which could be due to the error of the classifier or to the error of the user when labelling the samples. Note that the monitored plots have different planting dates, management practices, fertilization levels, and irrigation numbers, among other factors that may influence the results presented.

The quadratic SVM model trained and validated with the 86 extracted characteristics showed good performance in the classification of five phenological stages; however, it could not classify stage three (tassels and ears) because of the limited set of samples for this class or because the indicators used in the present study did not extract the characteristics of class three.

### 3.2. Adjustments to the Supervised Classification Models

To recognize five phenological stages of maize cultivation, through the 86 characteristics extracted from Sentinel 2 satellite images, a second training of the classifiers was performed. Table 6 shows the accuracy of the results obtained with the adjustments applied to the samples. The quadratic SVM model^1^ was the best classifier, with an overall accuracy of 81%, 7% higher than that before adjustment. The precision achieved by the classifier equalled the RF model proposed by Hejmanowska et al. [18] trained with the combination of Sentinel 1 and 2 data, slightly lower than the 81.85% achieved by the SVM model designed by Saini and Ghosh [20], higher than SVM (77.2%) [23]. The five classes were discriminated with accuracy greater than 70%.

Figure 5 shows the confusion matrix of the quadratic SVM model trained with the adjusted samples. It is observed that the precision by phenological stage slightly increased; 14% of the samples from stage one were classified as stage two (Figure 4), after adjustment (Figure 5) only 8% were discriminated as stage two; similarly, the error between stages five and four decreased by 2%. Therefore, it is inferred that the samples from stage three added noise to the model, decreasing the classification accuracy by class, and that some initial samples had been incorrectly labelled by the user; this could be due some of the digital photographs of the field being taken in areas where the development of the crop was not representative of the plot. With the 86 characteristics extracted, it was possible to train and validate the three classification models, and only the quadratic SVM confusion matrix is presented here because it achieved the best precision (Figure 5).

### 3.3. Determination of the Main Characteristics

Once it was demonstrated that with 86 characteristics, it was possible to recognize five phenological stages of maize crops, the characteristics were evaluated to determine which were the most representative, and the indicators that induced an increase in the accuracy of the classifiers were selected. Only 45 characteristics contributed to maximizing the accuracy of the implemented models, which are presented in Table 7.

Note that this is the first time that LISA has been implemented as an extractor of textural characteristics. The application of the method is inspired by the works of Appice and Malerba [15], who applied it for the recognition of plant cover.

The results of the LISA indicator express that it is a good descriptor of textural characteristics when the pixels of the extracted plots are grouped by the spatial dependence of eight neighbours and when applied to the RGB and NIR spectral bands and to the L*a*b* colour space, increasing the accuracy of the overall classification and that of each phenological stage of maize in the studied models.

The results obtained show that the LBP indicator is an extractor of texture characteristics that increases the accuracy of the classifier when applied to the RGB and NIR spectral bands and the Q band of the YIQ colour space, as has already been demonstrated in the recognition of crops [51,52], as well as in the classification of foliar diseases in grape cultivation [53] and recognition of leaves [54].

As shown in Table 6, the classifier that obtained the highest precision was the quadratic SVM, both for 86 characteristics and for 45, obtaining an accuracy of 81% for the first case and 82.3% for the second. The following section provides details of the trained classification model with 45 characteristics.

#### Impacts of the Indicators

To evaluate the impact of the indicators for the quadratic SVM model, six quadratic SVM classifiers were trained and validated by varying the set of main indicators. Figure 6 shows that the characteristics extracted by the RGB and NIR indicators had a 4% impact on the overall accuracy of the model (with overall accuracies of 45 and 37 features, respectively); additionally, the colour ratio indicator had a 3.5% impact, and LISA had a 1.9% impact. The characteristics extracted with the L*a*b*, LAI and LBP indicators generated impacts of 0.9, 0.7 and 0.6%, respectively, on the overall accuracy of the models.

The colour ratios, the RGB colour model, the NIR band and LISA extracted the most important characteristics for classifying the phenology of maize, which suggests that these ratios identify the intrinsic characteristics of the maize crop.

The proposed LISA indicator proved to be adequate for the extraction of textural features in satellite images since it led to a 1.9% increase in the overall accuracy of the quadratic SVM classifier, surpassing LBP, which accounted for only 0.6%. On the other hand, L*a*b* increased the accuracy by 0.9%, higher than the 0.7% of LAI; however, it was determined by analysis of variance with a significance level of 95% that the two indicators were equally favourable.

### 3.4. Training and Validation of the Supervised Classification Models with the Main Characteristics

To recognize the five phenological stages of maize cultivation through 45 characteristics extracted from Sentinel 2 satellite images, a third training and validation of the supervised classifiers with the most important extracted characteristics was performed. Table 6 shows the results of the accuracies by class and overall achieved with the trained classification models. Again, the quadratic SVM model stood out for all classes, with accuracies greater than 70% and an overall accuracy of 82.3%, close to the precision obtained by the SVM model (84.8%) proposed by Brinkhoff et al. [21] to discriminate perennial, annual, forest and other crops, slightly lower than the neural network (84.06%) [16], 5.16% lower than the SVM model designed by Ustuner et al. [55] to classify agricultural crops from data extracted from Rapideye satellite images with spatial resolution of 5 m and higher than the performance of the RF model (75%) proposed by Kushal et al. [56] to classify 4 classes of cover crops with data from Landsat 5, 7 and 8 images. Figure 7 shows the confusion matrix of the quadratic SVM model trained with the 45 most important characteristics. Phenological stages one (emergence) and six (end of senescence) were the best discriminated, with accuracies of 91 and 90%, respectively; class five (beginning of senescence) had the lowest classification accuracy (74%), confusing 17% with class four (formation and maturation of ears).

The accuracies reached by the quadratic SVM classifiers are similar to the results reported by Han et al. [29] in the classification of rice cultivation and to the convolutional neural network implemented by Yang et al. [30]. In both research studies, digital images taken with conventional cameras were processed with different spatial resolutions.

The results obtained in this research demonstrate the potential of Sentinel 2 images for the recognition of maize phenology and suggest that the indicators used in the present study are efficient in the extraction of the intrinsic characteristics of maize.

## 4. Conclusions

The supervised classification model that showed the best performance in the recognition of five phenological stages of maize cultivation from Sentinel 2 satellite images was the quadratic vector support machine; when trained with 86 characteristics, the overall accuracy was 81%, and when trained with 45 characteristics, the overall accuracy was 82.3%, where the class with the highest precision was the final stage of senescence (92%) and that with the lowest precision was the beginning of senescence (74%).

With the characteristics determined from the Sentinel 2 satellite images and the number of samples analysed, it was not possible to recognize the phenological stage called tassels and ears by any of the three classification models.

The RGB colour models and the NIR band, the colour ratios and LISA were the three indicators that extracted the most important intrinsic characteristics of maize crops; these indicators facilitated the classification of the phenological stages of the crop, impacting the general accuracy of the quadratic SVM model by 4.0, 3.5 and 1.9%, respectively.

The LISA indicator improved the accuracy of the classifiers when applied to the L*a*b* colour model and to the RGB and NIR bands.

The LBP method increased the general accuracy of the quadratic SVM model by 0.6% when applied to the RGB and NIR bands and the Q band of the YIQ colour space.

The increase in the number of characteristics used to train the supervised classification models did not provide greater precision in the classifiers, since by reducing the number of characteristics from 86 to 45, the precision obtained was similar. The results obtained suggest that the methodology proposed in the present research work is adequate for determining the five phenological stages of maize crops.

## Figures and Tables

**Figure 1 sensors-22-00094-f001:**
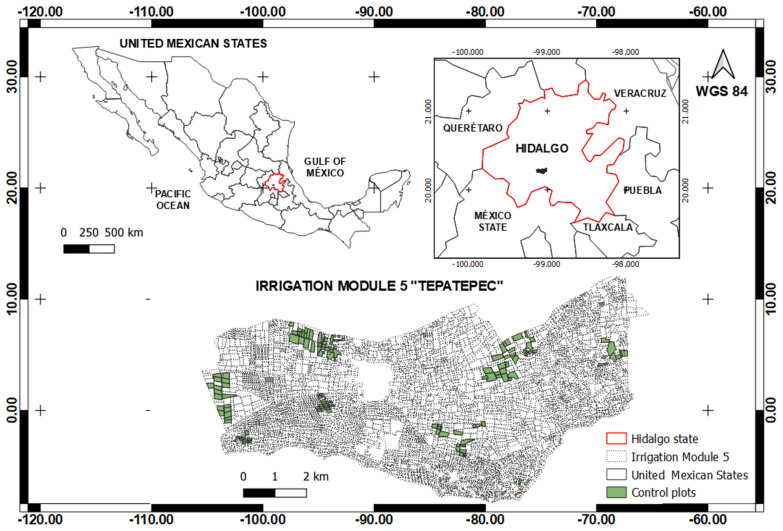
Localization of the study area.

**Figure 2 sensors-22-00094-f002:**
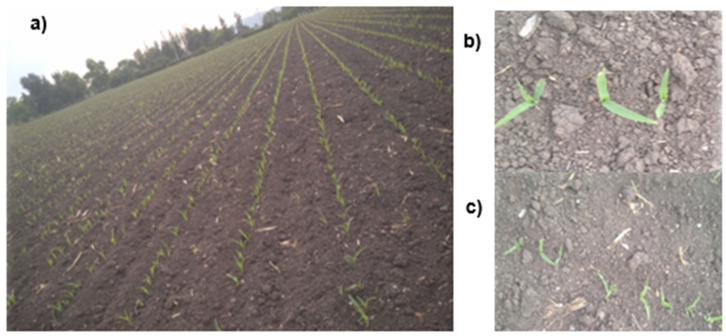
Taking photographs: (**a**) panoramic, (**b**) point and (**c**) overhead.

**Figure 3 sensors-22-00094-f003:**
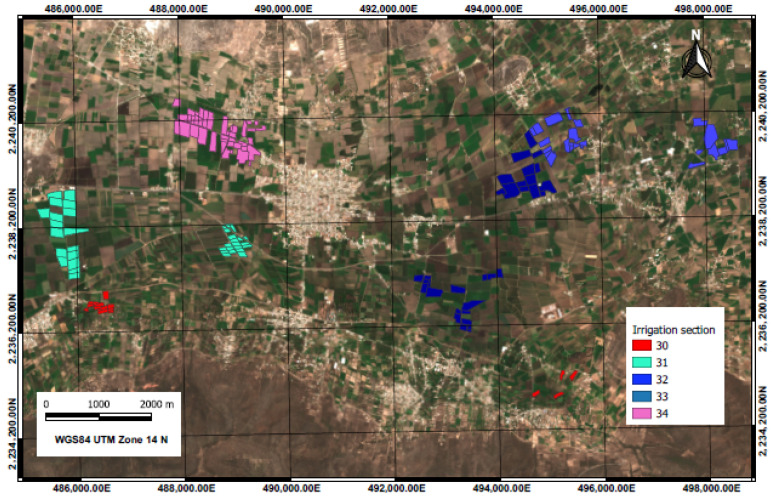
Map of sampling plots by irrigation section of the “Tepatepec” Irrigation Module in Irrigation District 03 “Tula”.

**Figure 4 sensors-22-00094-f004:**
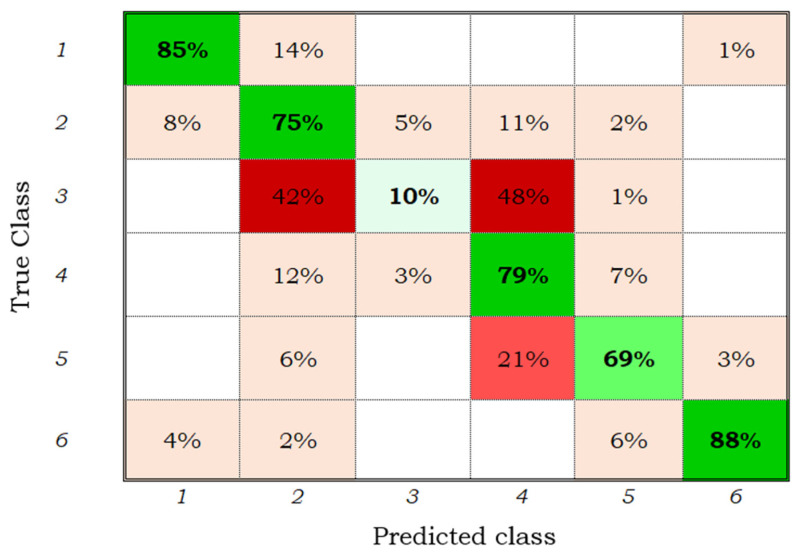
Confusion matrix of the quadratic SVM model for the six classes and 86 characteristics.

**Figure 5 sensors-22-00094-f005:**
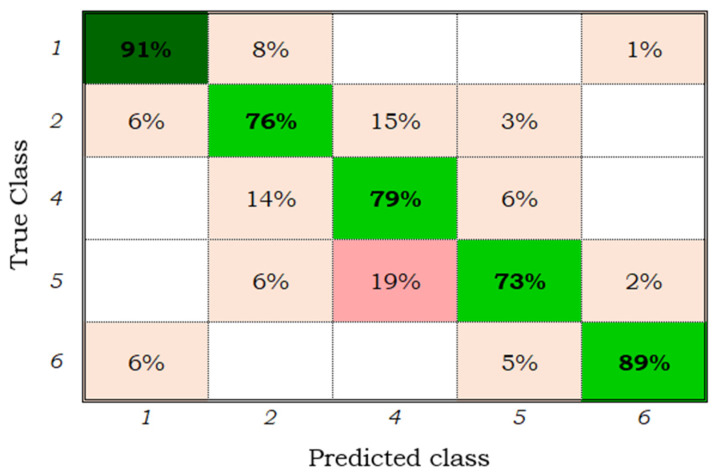
Confusion matrix of the quadratic SVM model for five classes and 86 characteristics.

**Figure 6 sensors-22-00094-f006:**
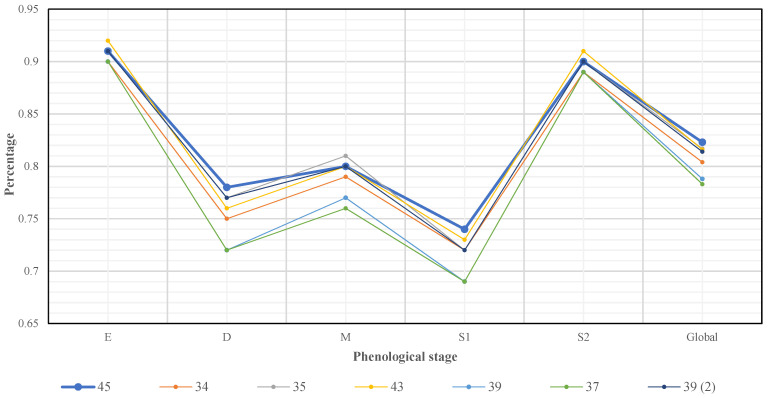
Evaluation of the main characteristics. In series 34, the LISA characteristics were eliminated, series 35 eliminated LBP, series 43 eliminated LAI, series 39 eliminated colour ratios, series 37 eliminated RGB and NIR, and series 39 (2) eliminated L*a*b*.

**Figure 7 sensors-22-00094-f007:**
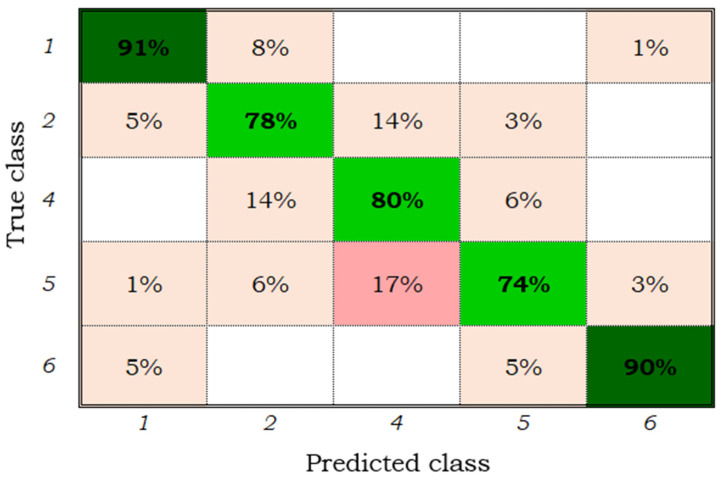
Confusion matrix of the quadratic SVM model for five classes and 45 characteristics.

**Table 1 sensors-22-00094-t001:** Characteristics of the Sentinel 2 satellite images.

Parameter	Description
Satellite:	Sentinel 2
Level:	2A
Used bands:	4 bands: B02, B03, B04 and B08
Number of images:	41 images
Data format:	uint 16
Dates:	03, 08, 13, 18, 23, 28 March 2019; 02, 07, 12, 17, 22, 27 April 2019; 02, 07, 12, 17, 22, 27 May 2019; 01, 06, 11, 16, 21, 26 June 2019; 06, 11, 16, 21, 31 July 2019; 05, 10, 15, 20, 25, 30 August 2019; 04, 19, 24 September 2019; 14, 24 October 2019; 13 November 2019
Cloud cover:	03 (0.03%), 08 (0.55%), 13 (0.03%), 18 (5.2%), 23 (1.34%), 28 (0.04%) March 2019; 02 (0.34%), 07 (0.02%), 12 (0.03), 17 (0.02%), 22 (0.07%), 27 (1.37%) April 2019; 02 (0.85%), 07 (0.04%), 12 (0.05%), 17 (0.93%), 22 (4.1%), 27 (7.15%) May 2019; 01 (7.82%), 06 (11.83%), 11 (11.49%), 16 (1.15%), 21 (8.83%), 26 (44%) June 2019; 06 (1.55%), 11 (19.83%), 16 (19.83%), 21 (7.81%), 31 (8.68%) July 2019; 05 (16.52), 10 (5.43%), 15 (3.09%), 20 (2.16%), 25 (7.78%), 30 (3.76%) August 2019; 04 (16.51%), 19 (23.05%), 24 (7.53%) September 2019; 14 (0.16%), 24 (3.74%) October 2019; 13 (9.76%) November 2019.

**Table 2 sensors-22-00094-t002:** Number of samples per class for the six phenological stages.

Stage Identifier	Stage	Number of Samples
1	Emergence (E)	715
2	Development (D)	1214
3	Tassels and ears (TE)	227
4	Formation and maturation of the ear (M)	963
5	Beginning of senescence (S1)	511
6	End of senescence (S2)	385

**Table 3 sensors-22-00094-t003:** Number of samples per class for five phenological stages.

Stage Identifier	Stage	Number of Samples
1	Emergence (E)	828
2	Development (D)	1085
4	Formation and maturation of the ear (M)	903
5	Beginning of senescence (S1)	520
6	End of senescence (S2)	428

**Table 4 sensors-22-00094-t004:** Set of evaluated descriptors.

Number of Features	Indicators	Indicator Removed
45	LISA, LBP, RGB, NIR, L*a*b*, YIQ, Colour ratios, LAI	-
34	LBP, RGB, NIR, L*a*b*, YIQ, Colour ratios, LAI	LISA
35	LISA, RGB, NIR, L*a*b*, YIQ, Colour ratios, LAI	LBP
43	LISA, LBP, RGB, NIR, L*a*b*, YIQ, Colour ratios	LAI
39	LISA, LBP, RGB, NIR, L*a*b*, YIQ, LAI	Colour indicators
37	LISA, LBP, L*a*b*, YIQ, Colour ratios, LAI	RGB and NIR
39	LISA, LBP, RGB, NIR, YIQ, Colour ratios, LAI	L*a*b*

**Table 5 sensors-22-00094-t005:** Accuracy of classifiers by class for six phenological stages.

Classification Model/Phenological Stage	E	D	TE	M	S1	S2	Global
LD	0.880	0.610	0.090	0.780	0.640	0.860	0.700
Quadratic SVM	0.850	0.750	0.100	0.790	0.690	0.880	0.744
kNN	0.790	0.580	0.100	0.630	0.510	0.790	0.614

The recorded accuracies correspond to the number of samples correctly classified of the total number of samples per phenological stage. E emergence, D development, TE tassels and ears, M formation and maturation of ears, S1 beginning of senescence and S2 end of senescence.

**Table 6 sensors-22-00094-t006:** Precision of classifiers by class for five phenological stages.

Classification Model/Phenological Stage	E	D	M	S1	S2	Global
LD ^1^	0.910	0.650	0.780	0.650	0.850	0.763
Quadratic SVM ^1^	0.910	0.760	0.790	0.730	0.890	0.810
kNN ^1^	0.880	0.650	0.660	0.510	0.800	0.699
LD ^2^	0.910	0.660	0.720	0.630	0.830	0.744
Quadratic SVM ^2^	0.910	0.780	0.800	0.740	0.90	0.823
kNN ^2^	0.880	0.640	0.670	0.540	0.810	0.705

^1^ Model trained with 86 characteristics to classify five phenological stages. ^2^ model trained with 45 characteristics to classify five phenological stages. The recorded accuracies correspond to the number of samples correctly classified among the total number of samples per phenological stage. E emergence, D development, TE tassels and ears, M formation and maturation of ears, S1 beginning of senescence and S2 end of senescence.

**Table 7 sensors-22-00094-t007:** Main characteristics.

Type	Indicator	Characteristics	Number of Features
Texture	LISA	lisa_rv, lisa_gv, lisa_bv, lisa_nirm, lisa_nirv, lmorl*_m, lmorl*_v, lmora*_m, lmora*_v, lmorb*_m, lmorb*_v	11
Texture	LBP	lbp_rm, lbp_rv, lbp_gm, lbp_gv, lbp_bm, lbp_bv, lbp_nirm, lbp_nirv, lbpq*_m, lbpq*_v	10
Colour	RGB and NIR	red_m, red_v, green_m, green_v, blue_m, blue_v, nir_m, nir_v	8
Colour	L*a*b*	l*_m, l*_v, a*_m, a*_v, b*_m, b*_v	6
Colour	YIQ	q*_m, q*_v	2
Vegetation	Colour indicators	Ratio_rm, Ratio_rv, Ratio_gm, Ratio_gv, Ratio_bm, Ration_bv	6
Vegetation	LAI	lai_m, lai_v	2

The final letter of each characteristic represents v: variance or m: mean of the pixels of the region of interest.

## Data Availability

The data presented in this study are available on request from the corresponding author.

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
