# Peer review of "Recognition of Maize Phenology in Sentinel Images with Machine Learning"

_sensors, 2021, doi:10.3390/s22010094_

Round 1

Reviewer 1 Report

Generaly manuscript is carefuly prepared and and is satisfactorily supported by recent references. There are some issues in methodology description and rationale which are indicated in pdf as comments and addressed. Hypothesis is not answered (confirmed/rejected) and consequently conclsion is not complete.  Technicaly, I guess the figures in manuscript should be translated to english and in better graphical quality, without fading edges.... but it is up to the editor.

Please take a look at the pdf for specific comments. 

Author Response

We are grateful to reviewer #1 for the encouraging words on our work, and for the comments and useful remarks that have helped us to improve our paper. We have taken your valuable suggestions into account in preparing the revised version.

Responses

It is unclear ..... What do you mean by "vegetation"? Can you describe clarify What kind of measure is it?

This observation was answered in section 2.4.3

It is not described how you are going to cofirm/reject your hypotesis.

This observation was answered in section 2.6.7

It is up to the editor if fading figure edges are convenient in this journal

The edge type of Figure 1 (now Figure 2) was changed.

If you describe naming scheme it should be translated

The observation was corrected.

It is unclear ..... Texture of what? What do you mean by "vegetation"? Can you describe clarify What kind of measure is it? Colour

A brief description was added on the characteristics of texture, colour and vegetation (lines 216-223).

What is rationale behind choosing L=0.5 ????

L is a factor that describes the density of the vegetation present in the soil, as the corn crop has a medium vegetation density, for this reason L = 0.5 was chosen, since 0 is for dense vegetation and 1 is low presence of vegetation.

Figure is graphicaly low quality. It is difficult to read it. Should be translated too.

Figure 4 was corrected

Figure is graphicaly low quality. It is difficult to read it. Should be translated too.

Figure 5 was corrected

I don't read here weather you or not confirmed your hypotesis.

A conclusion was added regarding the hypothesis of the work.

Best regards,

Antonia Macedo-Cruz

Alvaro Murguia-Cozar

Demetrio Fernandez Reynoso

 Jorge Salgado Transito

Reviewer 2 Report

With interest, I read the manuscript. It is appreciated that the manuscript is easy to follow and not too long. The message is clear and of interest to the community. The authors proposed a paper titled "Recognition of Maize Phenology in Sentinel Images with Machine Learning". The proposed paper seemed to be promising in terms of computational simplicity and classification accuracy. I would like to accept the manuscript in the present form.

Author Response

We are grateful to reviewer #2 for the encouraging words on our work, and for the comments and useful remarks that have helped us to improve our paper considerably.

 Best regards,

Antonia Macedo-Cruz

Alvaro Murguia-Cozar

Demetrio Fernandez Reynoso

 Jorge Salgado Transito

Reviewer 3 Report

1. section no since line 348 should be wrong which need re-arranged. 2. formulas in 2.4, 2.5. 2.6 which are copied could be replaced by further textual discription accustomed in crop context.

Author Response

Response to Reviewer 3

We are grateful to reviewer #3 for the encouraging words on our work, and for the comments and useful remarks that have helped us to improve our paper. We have taken your valuable suggestions into account in preparing the revised version.

Responses

  1. Section no since line 348 should be wrong which need re-arranged.

The concept "refinement" was changed to "debugging", which is more appropriate for the applied process. (Section 2.6.5)

Sections 2.6.1, 2.6.2 and 2.6.3 were renumbered 2.6.5, 2.6.6 and 2.6.7.

  1. formulas in 2.4, 2.5. 2.6 which are copied could be replaced by further textual discription accustomed in crop context.

We consider it important to keep the formulas in sections 2.4, 2.5, and 2.6 so that the reader identifies how the training data was calculated and our research can be replicated if required. 

Best regards,

 Antonia Macedo-Cruz

Alvaro Murguia-Cozar

Demetrio Fernandez Reynoso

 Jorge Salgado Transito

Reviewer 4 Report

Introduction  

To make the Introduction more substantial and to widen the issue of water resources management for agriculture, the authors may wish to provide more general and worldwide references on the issue.  

wider set of references relative to the remote sensing application for crop detection could be useful for the reader. Especially work more focused on crop identification rather than on more aggregated agricultural classes. In addition, limitation of the classification approaches for crop recognition by using only multispectral images could be highlighted.  

Materials and Methods  

More information about the set of satellite images used in the experiment would be beneficial for the reproducibility of the approaches described. In particular, a table listing the specific Sentinel product used and some metadata (e.g. cloud cover) might be necessary also as additional material. The table could be also accompanied by some indicator describing the density of images for each phenological stage.  

Information on any pre-processing of the Satellite images are missing as well as details about the level of the Sentinel-2 product used. 

figure with the geographical localization of the study area is missing.  

Figure 2 needs a legend in English.   

The significance of irrigation section is needed.  

As far as the definition of the indicators and classification models there is an unbalanced level of detail among those described. In particular, there is an excessive description for the Linear Discriminant Classifier that could be simplified by delineating the main characteristics and adding some more references on its application in similar contexts. The same apply to the SVM method where also two distinct figures are provided to exlain the characteristics of the method. Finally, the quadratic SVM is applied but this specific SVM implementation is not even cited in the SVM paragraph. 

Training and Validation 

The different number of samples available for each stage need a clarification. Is there any expected impact of the mentioned difference on the reliability of the results? 

Results and Discussions 

Figure 4, 5, 6 and 7 miss English labels. 

The discussion of the results, especially the accuracy achieved, need to be compared with other relevant studies addressing maize classification by using Sentinel 2 and more in general similar satellite images. 

Author Response

We are grateful to reviewer #4 for the encouraging words on our work, and for the comments and useful remarks that have helped us to improve our paper. We have taken your valuable suggestions into account in preparing the revised version.

 Responses

 Introduction  

To make the Introduction more substantial and to widen the issue of water resources management for agriculture, the authors may wish to provide more general and worldwide references on the issue.

R

A general review was carried out on the global problem of water management (lines 39-62)

A wider set of references relative to the remote sensing application for crop detection could be useful for the reader. Especially work more focused on crop identification rather than on more aggregated agricultural classes. In addition, limitation of the classification approaches for crop recognition by using only multispectral images could be highlighted.  

R

A review of the state of the art is carried out on the recognition of agricultural crops through sentinel 2 satellite images (lines 80-102).

The limitations of using multispectral images in the monitoring of crops were recorded (Lines 122-126).

Materials and Methods  

More information about the set of satellite images used in the experiment would be beneficial for the reproducibility of the approaches described. In particular, a table listing the specific Sentinel product used and some metadata (e.g. cloud cover) might be necessary also as additional material. The table could be also accompanied by some indicator describing the density of images for each phenological stage.  

R

Table 1 was added

Information on any pre-processing of the Satellite images are missing as well as details about the level of the Sentinel-2 product used. 

R

In lines 176-177 it is mentioned that for the present investigation no preprocessing was applied to the sentinel 2 images downloaded.

A figure with the geographical localization of the study area is missing.  

R

Figure 1 was added

Figure 2 needs a legend in English.   

R

Figure 2 (Figure 3, now) was corrected

The significance of irrigation section is needed.  

R

The plots to shape the “Tepatepec” irrigation module are dominated by a network of distribution channels. To ensure the proper functioning of the infrastructure, the total surface of the module is divided into five irrigation sections. This information is not relevant to the research work; however, it is mentioned in lines 203-206 so that Figure 3 is understandable.  

As far as the definition of the indicators and classification models there is an unbalanced level of detail among those described. In particular, there is an excessive description for the Linear Discriminant Classifier that could be simplified by delineating the main characteristics and adding some more references on its application in similar contexts. The same apply to the SVM method where also two distinct figures are provided to exlain the characteristics of the method. Finally, the quadratic SVM is applied but this specific SVM implementation is not even cited in the SVM paragraph. 

R

Section 2.6.1 linear discriminant classifier was resumed and references were added.

Section 2.6.3 Support Vector Machine was resumed, references were added.

Training and Validation 

The different number of samples available for each stage need a clarification. Is there any expected impact of the mentioned difference on the reliability of the results? 

R

The duration of the phenological stages varies, the satellite images of July, August, and September presented cloudiness over the control plots. This limited the number of samples extracted in some phenological stages, which is why the difference in samples. In the study work, the influence of the number of samples per phenological stage on the global precision of the models is not being evaluated, only the influence of the indicators and the number of characteristics extracted is being evaluated.

Results and Discussions 

Figure 4, 5, 6 and 7 miss English labels. 

R

Figures 4, 5, 6 y 7 were corrected

The discussion of the results, especially the accuracy achieved, need to be compared with other relevant studies addressing maize classification by using Sentinel 2 and more in general similar satellite images. 

R

The results obtained were compared with references related to the recognition of crops in satellite images (lines 487-490, 568-575).

Best regards,

Antonia Macedo-Cruz

Alvaro Murguia-Cozar

Demetrio Fernandez Reynoso

 Jorge Salgado Transito

Round 2

Reviewer 4 Report

I would suggest to improve Table 1. The mere list of images with their names is useless. this list can be more appropriate for a Supplementary materials content. An additional field with summary statistics on cloud cover could be useful.

Figure 1 can be improved. Toponyms are completely missing. The red outline is excessively thick. The two frames of the figure can be combined in a single picture by giving more space to the study area and less space to the overview map.

Author Response

We wish to thank you the reviewer #4, his comments and useful remarks that have helped us to improve our paper. We have taken your valuable suggestions into account in preparing the revised version.

Best regards,

Antonia Macedo-Cruz

Alvaro Murguia-Cozar

Demetrio Fernandez Reynoso

Jorge Salgado Transito
